# Arabic translation and cultural adaptation of sedentary behavior, dietary habits, and preclinical mobility limitation questionnaires: A cognitive interview study

Abdulrahman I. Alaqil [1,2,3]*, Nidhi Gupta[3], Shaima A. Alothman[4], Hazzaa M. Al-Hazzaa[4], Emmanuel Stamatakis[5‡], Borja del Pozo Cruz[2‡]

1 Department of Physical Education, College of Education, King Faisal University, Al-Ahsa, Saudi Arabia, 2 Center for Active and Healthy Ageing (CAHA), Department of Sports Science and Clinical Biomechanics, University of Southern Denmark, Odense, Denmark, 3 Department of Musculoskeletal Disorders and Physical Workload, National Research Centre for the Working Environment, Copenhagen, Denmark, 4 Lifestyle and Health Research Center, Health Sciences Research Center, Princess Nourah Bint Abdulrahman University, Riyadh, Saudi Arabia, 5 Charles Perkins Centre, School of Health Sciences, Faculty of Medicine and Health, The University of Sydney, Camperdown, New South Wales, Australia

‡ ES and BPC are joint senior authors on this work.
* Aialaqil@kfu.edu.sa

**Data Availability Statement:** Data cannot be shared publicly because of [confidentiality and privacy]. Data are available from the Institutional

## Abstract

### Background

Developing global evidence on the influence of health-related behaviors (e.g., sedentary behavior, diet) and mobility limitations on health requires global consortia from diverse sets of countries. Thus, the purpose was to translate and culturally adapt (i) the Sedentary Behavior Questionnaire (SBQ); (ii) the Dietary Habits Questionnaire adapted from the Survey of Health, Aging and Retirement in Europe (SHARE) study; (iii) the Preclinical Mobility Limitation questionnaire for use in the Saudi Arabian context.

### Method

50 adult Saudi participants (mean age 41.7±9.6, 48% female) participated in this study. We followed a systematic cross-cultural adaptation process that involved forward translation, synthesis, back-translation, expert panel, and pre-testing (cognitive interviewing). Four rounds of cognitive interviews were held with 40 participants for the SBQ, SHARE questionnaire, and the Preclinical Mobility Limitation questionnaire, an additional round was needed for the Preclinical Mobility Limitation questionnaire. Descriptive data (means ± standard deviations and frequencies with percentages) were reported for characteristics.

### Result

With some minor changes to the questionnaires, the SBQ, Dietary Habits, and Preclinical Mobility Limitation questionnaires were translated and cross-culturally adapted into Arabic. 100% of the participants confirmed that the resulting Arabic versions of the SBQ, Dietary Habits questionnaire, and Preclinical Mobility Limitation questionnaires were appropriate

Review Board (IRB), at Princess Nourah bin Abdulrahman University, Riyadh, KSA (contact via irb@pnu.edu.sa) for researchers who meet the criteria for access to confidential data.

**Funding:** Yes, This research was supported by the Deanship of Scientific Research, Vice Presidency for Graduate Studies and Scientific Research, King Faisal University, Saudi Arabia [Project No. GRANT2718]. The funders had no role in study design, data collection and analysis, the decision to publish, or the preparation of the manuscript.

**Competing interests:** The authors have declared that no competing interests exist.

and fully understandable for Arabic speakers in communicating the intended meanings of the items in each. For example, item SBQ1, 'Watching television (including videos on VCR/DVD)' was changed to 'Sitting and watching television or videos (including smartphones, tablets)'.

## Conclusion

The SBQ, Dietary Habits questionnaire, and Preclinical Mobility Limitation questionnaire were successfully cross-culturally adapted into Arabic and are now ready for use in Saudi Arabian.

## Introduction

Health-related behaviors, such as physical activity, sedentary behavior, and dietary habits are important for health, which may impact morbidity and mortality (i.e., cardiovascular disease, diabetes) [1–3]. Previous studies illustrate that sedentary behavior, poor diet, and a lack of mobility have all been linked to morbidity and mortality [1–6]. Measuring these risk factors remains critical for epidemiological and sub-clinical research to understand the long-term impacts of these behaviors on people and society.

Creating global evidence on how these risk factors influence health outcomes requires harmonizing and pooling data and creating knowledge on representative samples from different regions of the world [7–9]. One such effort is the Prospective Physical Activity, Sitting and Sleep consortium (ProPASS), a recent international research collaboration platform employing observational studies based on thigh-worn accelerometry devices [9]. One of the ultimate scientific objectives of ProPASS is "to develop methods for collecting data for future studies" and "to develop methods for processing, harmonizing, and pooling data of existing such studies" [10]. ProPASS employs various questionnaires that aim to assess the effect of physical activity, sedentary behavior, and sleep on a wide range of health outcomes in new cohorts that join ProPASS prospectively or retrospectively and attempts to standardize the measurements gathered by these cohort studies [10]. Recently, numerous large-scale epidemiological studies have joined ProPASS cohorts (e.g., the 1970 British Birth Cohort, the Australian Longitudinal Study on Women's Health, and the Trøndelag Health Study (HUNT) in Norway) and have used the questionnaires recommended by ProPASS [10]. However, although some of these recommended questionnaires by ProPASS are already available in Arabic, such as Global Physical Activity Questionnaire, some questionnaires are not available in Arabic (e.g., the Sedentary Behavior Questionnaire, the Dietary Habits questionnaire from the Survey of Health, Aging and Retirement in Europe, and Mobility Limitation Questionnaire). To address this, the authors' ultimate purpose in carrying out the present study is for it to be admitted to the ProPASS consortium as a prospective cohort study, making it the first prospective study from the Middle East to join the ProPASS. To ensure that the process of joining the consortium is scientifically sound, it is important to standardize the research instruments used to measure (subjectively or objectively) the above-mentioned health-related behaviors in the Middle East by ensuring that they are culturally appropriate and linguistically accurate for Saudi Arabian Arabic speakers.

Therefore, the authors selected these three research instruments thus that this study from Saudi Arabia can join ProPASS as these three research instruments are used by the consortium due to their reliability and validity. To measure sedentary behavior, the SBQ reports time spent sitting for nine activities during the week/weekend. The SBQ has been translated and

culturally adapted for use in various languages [11–13]. In addition, the adapted Dietary Habits questionnaire from the Survey of Health, Aging and Retirement in Europe (SHARE), and Mobility Limitation questionnaire have been used and translated into different languages [14, 15]. Previous research conducted on the Arabic population has highlighted some worrying trends, including high levels of sedentary behavior, unhealthy dietary habits, and limited mobility [16–19].

Furthermore, the need for translation comes from the fact that Arab-speaking countries constituted a large part of the world, and the Arabic language is the world's fourth most-spoken language [20]. However, still, not much evidence is created among these behaviors in Middle East region (especially in Saudi Arabia), and there is a scarcity of valid and reliable questionnaires for measuring sedentary behavior, dietary patterns, and mobility limitations in Arabic populations [17, 21–26]. Therefore, translating and culturally adapting these research instruments for use with Arabic-speaking populations including Saudi Arabia will contribute significantly to harmonizing and pooling data on the effects of these health-related behaviors. This will help the development of representative samples and provide a set of standardized questionnaires for future use to compare data from different populations and allow epidemiologists, policymakers, and researchers to exchange information across linguistic and cultural barriers [27, 28]. Therefore, this study aimed to translate and culturally adapt the Sedentary Behavior questionnaire (SBQ), Dietary Habits questionnaire (adapted from SHARE), and the Preclinical Mobility Limitation questionnaires in the Saudi Arabian context.

## Methods

### Study design and participants

A methodological cross-sectional study design was used to conduct the present study. This study employed a convenience sampling method to recruit adult Saudi participants ($n = 50$) aged >30 years old from Riyadh, Saudi Arabia. The participants were recruited from different locations across Riyadh using an online flyer (distributed via Twitter, and WhatsApp) as well as from among visitors to the Lifestyle and Health Research Center, Health Science Research Center at Princess Nourah bint Abdulrahman University, Riyadh, Saudi Arabia between May to June 2022. The study excluded participants with any medical conditions that may have compromised normal walking ability. This study was approved by the Institutional Review Board at Princess Nourah Bint Abdul Rahman University, Riyadh, Saudi Arabia (HAP-01-R-059) and was conducted according to the guidelines of the Declaration of Helsinki. Written informed consent was obtained from all participants. All participants voluntarily participated in the study and were informed of their right to withdraw at any time without stating a reason.

### Demographic and anthropometry measurement

The subjects' demographic data (e.g., sex, age, education level, nationality, gross income, marital status, smoking status, and employment status) was obtained by using a self-report form. In addition, each participant's weight was measured to the nearest 0.1 kg by using a portable SECA digital scale (Seca 813); height was measured to the nearest 0.1 cm by using a stadiometer (Seca 703 Secale). Waist circumference was measured midway between the lower rib margin and the iliac crest at the end of a gentle expiration to the nearest 0.1 cm [29]. Body mass index (BMI) was calculated as weight (Kg) divided by squared height ($m^2$). The anthropometry measurement outcomes were used for descriptive analysis.

### Testing procedures

After recruitment, the first author randomly divided the participants into 5 groups ($n = 10$) and contacted each participant to schedule a time for them to visit the Lifestyle and Health Research Center. After scheduling their visit, each participant in the first group was given a written consent form to sign and a demographic questionnaire to fill out; each participant's height, weight, and waist circumference were then measured. Next, participants filled out the sedentary behavior, dietary habits, and mobility questionnaires without assistance from the researcher. After completing the questionnaires, participants were cognitively interviewed by the first author. The same process was repeated with the other four groups.

### Questionnaires

The main author sought and obtained permission from Distinguished Professor Emeritus James F. Sallis to use and translate the SBQ. For the Dietary Habits (SHARE) questionnaire and Preclinical Mobility Limitation questionnaire were no need for permission to use or translated since these tools publicly opened through (https://www.share-datadocutool.org/control_construct_schemes/view/245) and (https://www.propassconsortium.org/).

This study details the process of translating and culturally adapting the three research instruments into Arabic for use with Arabic-speaking participants: the SBQ [30], the Dietary Habits (SHARE) questionnaire [31], and the Preclinical Mobility Limitation questionnaire [32]. (See S1 Appendix).

### Mobility questionnaire

The Preclinical Mobility Limitation questionnaire includes three items that had good construct validity and good internal consistency, with a Cronbach's alpha coefficient of 0.83 for identifying the early signs of disability and may be used as indicators to identify those at high risk of future disability [32]. These items are: *1) Do you have difficulty in walking 2.0km? 2) Do you have difficulty in walking 0.5km? and 3) Do you have difficulty in walking up 1 flight of stairs?* Each item is followed by five Likert-style responses: *(1) able to manage without difficulty, (2) able to manage with some difficulty, (3) able to manage with a great deal of difficulty, (4) able to manage only with help of another person, and (5) unable to manage even with help.*

### Dietary habits

Question items adapted from the Survey of Health, Aging and Retirement in Europe (SHARE) [31] were used to assess the dietary habits of the participants. The items are as follows: 1) Dairy products: *In the past month, how often do you have a serving of dairy products such as a glass of milk, cheese in a sandwich, a cup of yoghurt or a can of high-protein supplement?*, 2) Legumes and eggs: *In the past month, how often do you have a serving of legumes (200g), beans (3 tablespoons) or eggs (2 eggs/serving)?*, 3) Meat: *In the past month, how often do you eat meat, fish or poultry?*, and 4) Fruits and vegetables: *In the past month, how often do you consume a serving of fruits or vegetables?* The response categories for these questions are *Everyday -3-6 times a week —Twice a week—Once a week or—Less than once a week.* SHARE survey as a whole has been shown to be a valid and reliable tool for measuring a range of health-related variables, including those related to diet [31].

### Sedentary behavior questionnaire

The SBQ involves participants self-reporting the time they spent on sedentary behaviors. The SBQ covers nine sedentary behaviors (watching television, playing computer/video games,

sitting while listening to music, sitting and talking on the phone, doing paperwork or office work, sitting and reading, playing a musical instrument, doing arts and crafts, sitting and driving/riding in a car, bus, or train) [33]. The SBQ is considered to be a dependable tool for assessing sedentary behavior in adults, with reliability scores of 0.92 for total sedentary time. Furthermore, the SBQ has demonstrated a reasonable to substantial level of validity in measuring sedentary behavior among adults [33].

## Cross-cultural adaptation process

The current study followed the recommendations for cross-cultural adaptation studies by [34] (See Fig 1).

**1. Forward translation.** Two expert independent bilingual (English-Arabic) translators performed forward translation of the English versions of the three questionnaires. Translator 1 was very familiar with the concepts covered in the questionnaire. Translator 2 was neither aware nor informed of the concepts being quantified; this is in accordance with the guidance [34], which states that they should preferably have no background in the field.

**2. Synthesis of the translation.** The first author researcher compared the translated questionnaires produced during the initial stage and resolved any discrepancies. After that, the two translators and the authors produced a final agreed document for each questionnaire (the T12 questionnaire).

**3. Back translation.** The back translation of the T12 questionnaires was performed by two different bilingual independent translators. In this process, the two translators translated the T12 questionnaire version back into the original language (Arabic to English) without being aware of the original questionnaire or field. At this stage, back translation ensures that the questions are clear, understandable, and well translated. The key reason for this stage was to "avoid information bias and elicit unexpected meanings of the items in the translated questionnaire" (i.e., the T12 version).

**4. Expert committee.** The expert committee consisted of five specialists in the area of knowledge of the instruments and translators (forward and back translators). All the experts held PhD degrees, were bilingual (Arabic-English) and had significant experience in exercise physiology, physiotherapy, nutrition, public health, sports science, and language professional. The composition of the expert committee was vital to achieving satisfactory cross-cultural equivalence in the research instruments. The original questionnaires and each translation (forward and backward translation) together with the corresponding written reports composed by the first author were provided to each expert committee for review so they could make critical decisions and verify the cross-cultural equivalence of the original version and final versions [34].

**5. Pre-testing version.** The participants completed the questionnaires and underwent cognitive interviews to determine what they understood by the meaning of the items on each questionnaire and the respective chosen responses as defined below. The purpose of the pre-testing of the new questionnaire was to use the pre-final version on participants from the target setting.

## Cognitive interviews

A cognitive interview is a qualitative method specifically intended to scrutinize whether a survey question fulfills its intended purpose [35]. The cognitive interviews were conducted face-to-face separately after each participant had completed the final Arabic version of the translated questionnaires. The cognitive interviews consisted of four rounds for the SBQ and Dietary Habits questionnaire and five rounds for the Mobility Limitation Questionnaire; each

**Translation:**
- Two translators (T1 & T2)
- Into target language

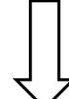

**Synthesis:**
- Synthesis T1 & T2 into T12 (Agreed document)
- Fix any discrepancies with translator's reports

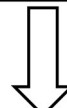

**Back translation:**
- Two translators mother tongue English (TB1 & TB2)
- Work from T-12 version

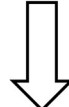

**Expert committee review:**
- Review all reports
- Committee consist of methodologist, language expert, translators, health expert
- Get an approval

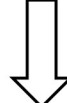

**Pretesting / cognitive interview**
- N=50 Adults
- Complete questionnaire
- Probe to get at understanding of item (cognitive interview)

**Fig 1. Flow diagram of cross-cultural adaptation recommended by Beaton, et al 2000.**

round involved ten participants (five male, five female). The cognitive interviews aimed to ensure that respondents could understand the questionnaire items and determine whether rewording or restructuring was required. All interview questions were semi-structured and open-ended (all questions were posed and answered in Arabic). The first author acted as an interviewer in this setting. The time spent answering the questions was recorded. During each round, the questions were modified based on the participants' feedback. The following details the questions that were asked to the participants during the cognitive interviews:

1. What do you think the question is about?

2. Is the question clear and understandable? If not, how can it be made clearer?

3. Do you have any questions about the items?

4. How could the wording be clearer?

   **Additional overall questions**:

1. Are there any activities or examples that we omitted?

2. Did any of the questions make you feel uncomfortable?

## Statistical analysis

Descriptive data analysis was performed. Continuous variables were described by using means ± standard deviations (SD), and the categorical variables were displayed by using frequencies with percentages (%). For the cognitive interviews, the overall total score of each round was calculated as follows: the number of responses (each question of the cognitive interview) × 100 ÷ total answers = % of responses that were not flagged as having issues with meaning). All data analyses were carried out using IBM's SPSS version 27 (version 27.0; SPSS, Inc. Chicago, Illinois).

## Results

### Cross-cultural adaptation

**Forward translation and synthesis of the translation.**   Working from the original questionnaire, two independent professional bilingual translators successfully translated the questionnaires into Arabic. Minor discrepancies were identified due to the literal translation of some English terms into Arabic. The authors and translators resolved these discrepancies during the initial translation phase, which resulted in the T12 version of the questionnaire.

**Backward translation.**   Two different independent bilingual translators translated the T12 version of the questionnaire from Arabic to English. This step functioned as a validity check however, minor modification was noted in the dietary habits questionnaire and have been resolved.

**Expert committee.**   The five experts reviewed the original questionnaires and the translated questionnaires together with the corresponding written reports provided by the authors. One of the experts withdrew due to family commitments. The expert committee provided advice on necessary cultural modifications to the questionnaire items. In the English version of the Dietary Habits Questionnaire, for example, the item related to fruit and vegetables contains some food items that are not consumed in Arabic countries This modification is recognized in the literature review [36].

**Cognitive interview.**   A total of 50 Saudi Arabian participants (48% female; 52% male) aged between 30 and 65 years old were interviewed. Four rounds of cognitive interviews were

performed with a total of 10 participants in each round for the SBQ, Dietary Habits questionnaire, and Preclinical Mobility Limitation questionnaire. An additional round of cognitive interviews was needed for the Preclinical Mobility Limitation questionnaire. Table 1. (below) summarizes the participant characteristics in each round.

The results of the cognitive interviews using the SBQ-Arabic version are presented in Table 2 (below). The overall participants' understanding of the intended meaning of the questionnaire items for rounds 1 and 2 was 98.89% and 100% for rounds 3 and 4. In addition, 100% of participants reported that the content of the SBQ-Arabic version was clear in all rounds. Meanwhile, the participants reported that the wording of the SBQ-Arabic version was

**Table 1. Summary characteristics of the sample population (n = 50) frequency (percentage).**

| Variables | R1(n = 10) | R2(n = 10) | R3(n = 10) | R4(n = 10) | R5(n = 10) |
|---|---|---|---|---|---|
| **Gender** | | | | | |
| Male | 6(60) | 4(40) | 6(60) | 5(50) | 5(50) |
| Female | 4(40) | 6(60) | 4(40) | 5(50) | 5(50) |
| **Marital Status** | | | | | |
| Single | 1(10) | 1(10) | 3(30) | | 4(40) |
| Married | 8(80) | 9(90) | 7(70) | 9(90) | 6(60) |
| Divorced | 1(10) | | | 1(10) | |
| **Education** | | | | | |
| Middle School | 1(10) | 1(10) | | 2(20) | |
| High School | 2(20) | 3(30) | 2(20 | | 5(50) |
| College degree | 2(20) | 3(30) | 2(20) | 6(60) | 3(30) |
| Post-graduate degree | 5(50) | 3(30) | 6(60) | 2(20) | 2(20) |
| **Income (Saudi Riyal)** | | | | | |
| 10000 or less | 5(50) | 5(50) | 3(30) | 5(50) | 5(50) |
| 10001 to 20000 | 3(30) | 2(20) | 2(20) | 4(40) | 5(50) |
| 20001 to 30000 | 2(20) | 3(30) | 4(40) | 1(10) | |
| 30001 and more | | | 1(10) | | |
| **Employment status** | | | | | |
| Employed | 7(70) | 6(60) | 10(100) | 9(90) | 9(90) |
| Student | | 1(10) | | | |
| Homemaker | | | | 1(10) | 1(10) |
| Unpaid voluntary work | | 1(10) | | | |
| Retired | 3(30) | 2(20) | | | |
| **Work type** | | | | | |
| Desk Job | 4(40) | 1(10) | 5(50) | 5(50) | 5(50) |
| Physical | | 2(20) | | | 1(10) |
| Mixed | 5(50) | 3(30) | 5(50) | 4(40) | 3(30) |
| **Mean ± SD** | | | | | |
| | R1 | R2 | R3 | R4 | R5 |
| Age (Years) | 39±7 | 48±12 | 39.9±7.1 | 42±8.8 | 35±5.4 |
| Weight (kg) | 79.3±11 | 78.5±8.3 | 79.3±11 | 86.4±18.8 | 83.6±14.5 |
| Height (cm) | 1.68±.08 | 1.63±.09 | 1.68±.07 | 1.66±.08 | 1.69±.09 |
| Waist circumference (cm) | 100±3.2 | 103±11.7 | 100±3.2 | 108±13.5 | 99±7.8 |
| Body Mass Index (kg·m-2) | 26.7±2.5 | 29.7±3.9 | 27.7±2.5 | 31.2±5.5 | 29±3.7 |

**R** = round, **SD** = standard deviation

**Table 2. Overall result of cognitive interview of the sedentary behavior, dietary habits, and mobility limitation questionnaires Arabic version.**

| Rounds | n | Participant Understanding of the Intended Meaning | The Content Was Clear for the Participant | The Wording Was Clear for the Participant |
|---|---|---|---|---|
| **Sedentary Behavior Questionnaire** | | | | |
| R1 | 10 | 98.89% | 100% | 81.12% |
| R2 | 10 | 98.89% | 100% | 95.6% |
| R3 | 10 | 100% | 100% | 98.89% |
| R4 | 10 | 100% | 100% | 100% |
| **Dietary Habits Questionnaire** | | | | |
| R1 | 10 | 100% | 100% | 97.5% |
| R2 | 10 | 100% | 100% | 95% |
| R3 | 10 | 100% | 100% | 95% |
| R4 | 10 | 100% | 100% | 100% |
| **Mobility Limitation questionnaire** | | | | |
| R1 | 10 | 93.3% | 100% | 86.6% |
| R2 | 10 | 93.3% | 100% | 80% |
| R3 | 10 | 96.6% | 100% | 83.3% |
| R4 | 10 | 100% | 100% | 86.6% |
| R5 | 10 | 100% | 100% | 100% |

clear in round 1 (81.12%), round 2 (95.6%), round 3 (98,89%), and round 4 (100%), respectively.

During the process, some changes to the first draft were required to improve participants' understanding; thus, item SBQ1 '*Watching television (including videos on VCR/DVD)*' was changed to '*Sitting and watching television or videos (including smartphones, tablets)*'; item SBQ2 '*Playing computer or video games*' was modified to '*Sitting and playing computer or video games*'; item SBQ3 '*Sitting and listening to music on the radio, tapes, or CDs*' was changed to '*Sitting and listening to podcasts (including the Qur'an, duea, news, e-books, or music) on the radio, smartphones, tablets*'; item SBQ6 '*Sitting and reading a book or magazine*' was changed to '*Sitting and reading a book or magazine or on smartphones and tablets*'; item SBQ7 '*Playing a musical instrument*' was modified to '*Sitting and playing a musical instrument*'; item SBQ8 '*Doing artwork or crafts*' was modified to '*Sitting and doing artwork or crafts*'. After the modifications, the questionnaire was then given to the participants again; this time, no difficulties were reported in understanding the questionnaire items.

The overall results demonstrated that all participants (100%) understood the intended meanings of the items on the Arabic version of the Dietary Habits questionnaire in all rounds. Also, all participants (100%) reported that the content of the Dietary Habits questionnaire was clear for them in all rounds. However, the percentages of participants who declared that the wording of the items on the Dietary Habits questionnaire was clear for them were as follows: round 1 (97.5%), round 2 (95%), round 3 (95%), and round 4 (100%), respectively. (See Table 2).

In addition, the Dietary Habits questionnaire required one minor change to improve the participants' understanding of item 2: '*In the past month, how often do you have a serving of legumes (200g), beans (3 tablespoons) or eggs (2 eggs/serving)*?' was changed to '*In the past month, how often do you have a serving of legumes (14 tablespoons), beans (3 tablespoons) or eggs (2 eggs/serving)*?' due to the greater popularity of this means of measurement among the Saudi participants.

A total of 50 participants participated in the cognitive interviews using the Arabic version of the Preclinical Mobility Limitation questionnaire; the results are presented in Table 2. The

interviews consisted of five rounds, the overall results of the first and second rounds demonstrated that 93.3% of the participants understood the intended meaning of the questionnaire items, and 100% reported that the content was clear to them. In contrast, 86.6%, and 80% of participants reported that the questionnaire items were semantically clear for them. In round 3 some modifications were required to make the questionnaire clearer for the participants. Therefore, we added an example to items 1 and 2; however, our example for this item contained the word "intensity" and we noted some of the participants did not report any problem with understanding the meaning of the item. However, the expert committee recommended removing the word "intensity" (please see below) and recruiting more participants to ensure this questionnaire was well understood. In rounds 3–5, 100% of the participants reported understanding the intended meaning of the questionnaire items, 100% reported that the content was clear for them, and 100% reported that the items were semantically clear.

Furthermore, some modifications to the Mobility Limitation questionnaire were required to improve the participants' understanding; thus, item 1, *'Do you have difficulty in walking 2.0km?'* was changed to *'Do you have difficulty in walking 2.0km? (Which is equivalent to walking for 20–25 minutes)'*; item 2, *'Do you have difficulty in walking for half a kilometer (500 meters)'* was changed to *'Do you have difficulty in walking for half a kilometer (500 meters)? (Which is equivalent to walking for less than 10 minutes)'*. Following the adjustments, the questionnaire was administered again to the participants, who reported no issues with comprehending its contents. The participants did not indicate that any activities or instances were missed, and they also did not express discomfort in response to any of the questions.

## Discussion

Most of the available research instruments for measuring health-related behaviors have been developed in English-speaking countries. This prevents the participation of Saudi Arabia in ProPASS at present because these research instruments can only be used with English-speaking populations due to language and cultural barriers [32–34]. To overcome this and enable Saudi Arabia to join ProPASS, the current study aimed to cross-culturally adapt, translate into Arabic, and evaluate the SBQ, Dietary Habits questionnaire, and the Preclinical Mobility Limitation questionnaire for use in Saudi Arabia to enable it to join ProPASS.

The research instruments were translated and culturally adapted to ensure equivalence between the original versions and the versions produced as part of this study [34]. The SBQ, Dietary Habits questionnaire and the Preclinical Mobility Limitation questionnaires have been translated into many languages (i.e., Turkish, Spanish, Korean, German, Danish) and have been used across many different ethnic groups [12, 31, 37]. The availability of an Arabic-language version of the Sedentary Behavior Questionnaire that is valid for use with Arabic-speaking populations is crucial for research into lifestyle behaviors and non-communicable diseases in Middle Eastern countries due to the high prevalence of physical inactivity and sedentary behavior in such regions [21, 22, 25]. Due to the lack of a suitable research instrument in Arabic to measure engagement in sedentary behaviors during weekdays and weekends [19], we translated and culturally adapted the Sedentary Behavior Questionnaire into Arabic to make the questionnaire items fully comprehensible to Arabic-speaking participants. In summary, the results show that the SBQ, Dietary Habits questionnaire, and Preclinical Mobility Limitation questionnaire were successfully translated into Arabic and culturally adapted for use in the Saudi Arabia context. This means that these three Arabic-language research instruments can now be used in the consortia setting to make valuable contributions to global ProPASS data. Since the ProPASS Saudi Arabia project is an extension of the ProPASS consortium, which has a comprehensive scope that includes physical activity, sedentary behavior, and

sleep, as well as other health-related areas such as dietary habits and mobility limitations. The ProPASS consortium aims to generate evidence-based recommendations to improve public health in these various domains [9].

Furthermore, the Dietary Habits questionnaire was translated and culturally adapted into Arabic with minor changes to ensure it was more closely related to Arab contexts in terms of food-consumption patterns and measurements. In Eastern Mediterranean countries, food-consumption patterns differ dramatically compared to western countries [16, 36]; therefore, during the cross-cultural adaptation process, we modified some of the food-related examples and measuring units to make them more comprehensible and appropriate for Middle Eastern participants. Our adaptation aligned with a prior investigation conducted by Al-Farhan, et al in 2021, in which they made modifications to certain food options based on religious limitations and removed others that were either unavailable, not popular, or not commonly consumed among the Arab population [38]. It is hoped that the Arabic version of the Dietary Habits questionnaire will help researchers, public health officials, and nutritionists, in Arabic-speaking countries to achieve an appropriate estimation of the dietary habits of Arab populations.

The Preclinical Mobility Limitation Questionnaire is a crucial tool for predicting the risk of future disability [32]. Three questions were translated and culturally adapted into Arabic; however, some difficulty was noted with the participants' understanding of these items, especially Q1 and Q2. During the cognitive interviews, the participants reported that it was difficult to estimate the time that walking two kilometers and 500 meters, respectively, would take. To overcome this, we added an estimation to the items of the times required to walk the above distances to make these items clearer and better understood. After all these moderations were completed, the questionnaire is now considered ready for use with Arabic-speaking populations.

## Study strengths and limitations

This study has two main strengths. First, it provides culturally adapted Arabic translations of the SBQ, Dietary Habits questionnaire, and the Preclinical Mobility Limitation questionnaire so that they can be widely used to assess physical behaviors in Arabic-speaking populations. Second, the translation and cultural validation process involved an adequate sample size for the cognitive interviews (50 participants), and the pre-testing sample size was adequate according to cross-cultural adaptation guidelines [34]. Despite its merits, this study is subject to two limitations. First, it does not include participants <30 years old or >65; this is because these age ranges are not required by ProPASS. Second, it does not provide an examination of the psychometric properties of the current questionnaires; however, future work is expected to cover this area.

## Conclusion

The SBQ, Dietary Habits questionnaire, and Preclinical Mobility Limitation questionnaire were successfully translated and culturally adapted for use with Arabic-speaking populations. These three research instruments will facilitate the collection of health data among Saudi Arabian and other Arabic-speaking populations for use with large global health initiatives such as ProPASS [10]. Future studies are needed to further test the psychometric properties of these research instruments, including their test-retest reliability and validity in the Arabic context.

## Supporting information

**S1 Appendix.**
(DOCX)

## Acknowledgments

The authors would like to express their gratitude to all participants, translators, and the expert committee for their contributions to the cross-cultural process.

## Author Contributions

**Conceptualization:** Abdulrahman I. Alaqil, Shaima A. Alothman, Hazzaa M. Al-Hazzaa, Emmanuel Stamatakis, Borja del Pozo Cruz.

**Data curation:** Abdulrahman I. Alaqil, Shaima A. Alothman, Hazzaa M. Al-Hazzaa.

**Formal analysis:** Abdulrahman I. Alaqil, Borja del Pozo Cruz.

**Funding acquisition:** Abdulrahman I. Alaqil.

**Investigation:** Abdulrahman I. Alaqil, Shaima A. Alothman, Hazzaa M. Al-Hazzaa, Emmanuel Stamatakis, Borja del Pozo Cruz.

**Methodology:** Abdulrahman I. Alaqil, Nidhi Gupta, Shaima A. Alothman, Hazzaa M. Al-Hazzaa, Emmanuel Stamatakis, Borja del Pozo Cruz.

**Project administration:** Abdulrahman I. Alaqil, Nidhi Gupta, Emmanuel Stamatakis.

**Supervision:** Abdulrahman I. Alaqil, Nidhi Gupta, Emmanuel Stamatakis, Borja del Pozo Cruz.

**Writing – original draft:** Abdulrahman I. Alaqil, Shaima A. Alothman, Hazzaa M. Al-Hazzaa, Borja del Pozo Cruz.

**Writing – review & editing:** Abdulrahman I. Alaqil, Nidhi Gupta, Shaima A. Alothman, Hazzaa M. Al-Hazzaa, Emmanuel Stamatakis, Borja del Pozo Cruz.

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
