## [Decision Letter · Decision Letter 0]

22 Feb 2023

PONE-D-22-32960Sedentary Behavior, Dietary Habits, and Preclinical Mobility Limitation Questionnaires: Translation and Cross-Cultural Adaptation to Arabic PopulationPLOS ONE

Dear Dr. Abdulrahman 

Thank you for submitting your manuscript to PLOS ONE. After careful consideration, we feel that it has merit but does not fully meet PLOS ONE’s publication criteria as it currently stands. Therefore, we invite you to submit a revised version of the manuscript that addresses the points raised during the review process.

We look forward to receiving your revised manuscript.

Kind regards,

Sally Mohammed Farghaly

Academic Editor

PLOS ONE

Journal Requirements:

"In addition, the authors would like to thank the Deanship of Scientific Research, Vice Presidency for Graduate Studies and Scientific Research, King Faisal University, Saudi Arabia for the financial support under Ambitious Researcher track [Project No. GRANT1616]. "

"Yes, This research was supported by the Deanship of Scientific Research, Vice Presidency for Graduate Studies and Scientific Research, King Faisal University, Saudi Arabia [Project No. GRANT1616]."

"Yes, This research was supported by the Deanship of Scientific Research, Vice Presidency for Graduate Studies and Scientific Research, King Faisal University, Saudi Arabia [Project No. GRANT1616]."  

Reviewers' comments:

Reviewer's Responses to Questions

**Comments to the Author**

1. Is the manuscript technically sound, and do the data support the conclusions?

Reviewer #1: Yes

Reviewer #2: Yes

2. Has the statistical analysis been performed appropriately and rigorously? 

Reviewer #1: Yes

Reviewer #2: Yes

3. Have the authors made all data underlying the findings in their manuscript fully available?

Reviewer #1: Yes

Reviewer #2: Yes

4. Is the manuscript presented in an intelligible fashion and written in standard English?

Reviewer #1: Yes

Reviewer #2: Yes

5. Review Comments to the Author

Reviewer #1: Thank you so much for choosing me to review this manuscript.

Sedentary Behavior, Dietary Habits, and Preclinical Mobility Limitation Questionnaires: Translation and Cross-Cultural Adaptation to Arabic Population

The title needs to be changed.

The author uses the cognitive interviewing method to translate the study tools. So the title will be

Arabic Translation and Cross-Cultural Adaptation of Population Sedentary Behavior, Dietary Habits, and Preclinical Mobility Limitation Questionnaires: A cognitive interview study.

The abstract is excellent and does not need any modifications

Introduction

Thank you so much for the made an effort in this section to the point of comprehensive

I need to add a paragraph about the importance of the tools used in the study from the perspectives of their specialties, especially the Musculoskeletal Disorders and Physical Workload specialty and the Center for Active and Healthy Aging.

Line 83, when you illustrate the SBQ, the information in this section was repeated in the sections of tool description; modify or rephrase it without repetition.

Mobility questionnaire section

Add a reliability test used for every tool with reference at the end of every description of the tool (The Preclinical Mobility Limitation questionnaire, Dietary Habits, and Sedentary Behavior Questionnaire)

Results in line 285 for round 1 percentage are 81.12%; modify it.

Why did you add a round of cognitive interviews to the preclinical mobility limitation questionnaire?

Other two tools, the authors just use 4 rounds. Why in this tool do they use 5 rounds; can you write the reason?

Is there any limitation in this study?

References 14 and 35 were repeated.

References 6 and 7 were repeated.

Cheers

Reviewer #2: Title: Sedentary Behavior, Dietary Habits, and Preclinical Mobility Limitation Questionnaires: Translation and Cross-Cultural Adaptation to Arabic Population”

Reviewer's report:

Thanks for allowing me to review your interesting manuscript titled " Sedentary Behavior, Dietary Habits, and Preclinical Mobility Limitation Questionnaires: Translation and Cross-Cultural Adaptation to Arabic Population”

Dear Authors,

kindly find the following points that need for your careful review and work on your research

article quality improvement:

Introduction:

- ProPASS: This is in need for operational definition by the meaning and the reason for selected that area for data collection.

- Theoretical background regarding Sedentary Behavior, Dietary Habits, and Preclinical Mobility Limitation current situation in Arabic population to be expanded in the introduction section. It is recommended to clarify the theoretical background of your research variables based on the aim the research.

- The research question: not found

Methods:

- Complete research design is required to be clarified in details

- The sample calculation and the technique of sampling is required to be clarified.

- This word (researcher) should be replaced by the authors or researchers

- pilot sample not clear.

- The tools reliability and validity tests need to be clarified.

- please mention the exact time of sampling when began and ended.

Results:

- Table (1) needed to revised because percentage with frequency not correct.

- The tables 2,3,4 can be collected in one table and reorganized and what about other questions in cognitive interview such as, are there any activities or examples that we omitted?

Discussion:

- The discussion section should align by the results section and add interpretation about the

findings and support that interpretation with the different international studies, but actually it

miss that and needs for improvement.

-Limitation of study should be in separated section.

- Implication of study should be write.

6. PLOS authors have the option to publish the peer review history of their article (what does this mean?). If published, this will include your full peer review and any attached files.

Reviewer #1: **Yes: **Ayman Mohamed El-Ashry, lecturer of psychiatric and mental health nursing, faculty of nursing, Alexandria university, Egypt.

Reviewer #2: **Yes: **Boshra Karem Mohamed Elsayed

---

## [Author Response · Author response to Decision Letter 0]

10 Mar 2023

Point-by-Point Reviewer Comments:

Reviewer 1

Thank you so much for choosing me to review this manuscript.

Sedentary Behavior, Dietary Habits, and Preclinical Mobility Limitation Questionnaires: Translation and Cross-Cultural Adaptation to Arabic Population

The title needs to be changed.

The author uses the cognitive interviewing method to translate the study tools. So, the title will be

Arabic Translation and Cross-Cultural Adaptation of Population Sedentary Behavior, Dietary Habits, and Preclinical Mobility Limitation Questionnaires: A cognitive interview study.

Thank you for your comments and we have changed title to 

“Arabic Translation and Cultural Adaptation of Sedentary Behavior, Dietary Habits, and Preclinical Mobility Limitation Questionnaires: A Cognitive Interview Study”

The abstract is excellent and does not need any modifications

Thank you!

Introduction

Thank you so much for the made an effort in this section to the point of comprehensive

I need to add a paragraph about the importance of the tools used in the study from the perspectives of their specialties, especially the Musculoskeletal Disorders and Physical Workload specialty and the Center for Active and Healthy Aging.

We appreciate your feedback, and we believe that the information we provided about ProPASS was useful and thorough. However, we apologize that we couldn't fully comprehend your comments, which is why we couldn't incorporate additional information into our revisions. Nonetheless, we hope that the revised version meets your expectations.

Line 83, when you illustrate the SBQ, the information in this section was repeated in the sections of tool description; modify or rephrase it without repetition.

Thank you for your comments and this sentence has been revised

 “The SBQ reports time spent sitting for nine activities during the week/weekend”

Mobility questionnaire section

Add a reliability test used for every tool with reference at the end of every description of the tool (The Preclinical Mobility Limitation questionnaire, Dietary Habits, and Sedentary Behavior Questionnaire)

Thank you for your comments, we added the reliability for each tool, please see the following information in the manuscript: 

“SHARE survey as a whole has been shown to be a valid and reliable tool for measuring a range of health-related variables, including those related to diet (29).”

Results in line 285 for round 1 percentage are 81.12%; modify it.

Thank you for your comments, we Rectified. 

Why did you add a round of cognitive interviews to the preclinical mobility limitation questionnaire?

Other two tools, the authors just use 4 rounds. Why in this tool do they use 5 rounds; can you write the reason?

Thank you for your comments, we added one more round of interview to preclinical mobility limitation due to about 15% of participants had problem with “The wording was clear for participants” thus, a fifth round of interview was necessary. Willis, G. B. (2015) 

Is there any limitation in this study?

Thank you for your comments, we already included the “strength and limitations of the study” in the last paragraph as separate section 

“This study has two main strengths. First, it provides culturally adapted Arabic translations of the SBQ, Dietary Habits questionnaire, and the Preclinical Mobility Limitation questionnaire so that they can be widely used to assess physical behaviors in Arabic-speaking populations. Second, the translation and cultural validation process involved an adequate sample size for the cognitive interviews (50 participants), and the pre-testing sample size was adequate according to cross-cultural adaptation guidelines (32). Despite its merits, this study is subject to two limitations. First, it does not include participants <30 years old or >65; this is because these age ranges are not required by ProPASS. Second, it does not provide an examination of the psychometric properties of the current questionnaires; however, future work is expected to cover this area. 

References 14 and 35 were repeated.

Thank you for your comment we checked it.

References 6 and 7 were repeated.

Thank you for your comment we checked it.

Cheers

 

Reviewer 2

Thanks for allowing me to review your interesting manuscript titled " Sedentary Behavior, Dietary Habits, and Preclinical Mobility Limitation Questionnaires: Translation and Cross-Cultural Adaptation to Arabic Population”

Dear Authors,

kindly find the following points that need for your careful review and work on your research

article quality improvement:

We hope you find the revisions adequate

Introduction:

- ProPASS: This is in need for operational definition by the meaning and the reason for selected that area for data collection.

We have clarified and provided further elaboration for this section as described below and in the manuscript:

“ProPASS employs various questionnaires that aim to assess the effect of physical activity, sedentary behavior, and sleep on a wide range of health outcomes in new cohorts that join ProPASS prospectively or retrospectively and attempts to harmonize the measurements gathered by these cohort studies (10).”

-Theoretical background regarding Sedentary Behavior, Dietary Habits, and Preclinical Mobility Limitation current situation in Arabic population to be expanded in the introduction section. It is recommended to clarify the theoretical background of your research variables based on the aim the research.

Thank you for your comments, we have added the following information into the manuscript: 

“Previous research conducted on the Arabic population has highlighted some worrying trends, including high levels of sedentary behavior, unhealthy dietary habits, and limited mobility (16–19).”

- The research question: not found

Thank you for your comment, you could find it in lines 107 – 110 in the manuscript:

“Therefore, this study aimed to translate and culturally adapt the Sedentary Behavior questionnaire (SBQ), Dietary Habits questionnaire (adapted from SHARE), and the Preclinical Mobility Limitation questionnaires in the Saudi Arabian context.”

Methods:

- Complete research design is required to be clarified in details

We have added the following information into the Study design and participants section:

“A methodological cross-sectional study design was used to conduct the present study.”

- The sample calculation and the technique of sampling is required to be clarified.

We appreciate your comment. In accordance with the recommendations from Beaton et al.'s 2000 study, our research has adhered to a sample size of 30 to 40 participants for cross-cultural studies to complete the necessary tests. This guideline has been followed in our study.

- This word (researcher) should be replaced by the authors or researchers

Thank you for your comment and We have modified the word “researcher” to “authors”

- pilot sample not clear.

Thank you for your comments and we followed the sample size (30 – 40) participants that has recommended by Beaton, et al 2000. 

- The tools’ reliability and validity tests need to be clarified.

Thank you for your comments and we added the reliability for each tool please see the following information that included in manuscript: 

 “The Preclinical Mobility Limitation questionnaire includes three items that had good construct validity and good internal consistency, with a Cronbach's alpha coefficient of 0.83 for identifying the early signs of disability and may be used as indicators to identify those at high risk of future disability (32)”

“SHARE survey as a whole has been shown to be a valid and reliable tool for measuring a range of health-related variables, including those related to diet (31)”

“The SBQ is considered to be a dependable tool for assessing sedentary behavior in adults, with reliability scores of 0.92 for total sedentary time. Furthermore, the SBQ has demonstrated a reasonable to substantial level of validity in measuring sedentary behavior among adults(33).”

- please mention the exact time of sampling when began and ended.

Thank you for your comments and we added the started and ended month

“The participants were recruited from different locations across Riyadh using an online flyer (distributed via Twitter, and WhatsApp) as well as from among visitors to the Lifestyle and Health Research Center, Health Science Research Center at Princess Nourah bint Abdulrahman University, Riyadh, Saudi Arabia between May to June 2022.”

Results:

- Table (1) needed to revised because percentage with frequency not correct.

This table has been revised. 

- The tables 2,3,4 can be collected in one table and reorganized and what about other questions in cognitive interview such as, are there any activities or examples that we omitted?

Thank you for your comment. We did not receive any responses from participants, and we have included the results of the additional question in lines 350-351:

“The participants did not indicate that any activities or instances were missed, and they also did not express discomfort in response to any of the questions.”

Discussion:

- The discussion section should align by the results section and add interpretation about the findings and support that interpretation with the different international studies, but actually it miss that and needs for improvement.

Thank you for your comment, we have clarified it please see the following info in the manuscript:

“Since the ProPASS Saudi Arabia project is an extension of the ProPASS consortium, which has a comprehensive scope that includes physical activity, sedentary behavior, and sleep, as well as other health-related areas such as dietary habits and mobility limitations. The ProPASS consortium aims to generate evidence-based recommendations to improve public health in these various domains (39).”

In line 393 “Our adaptation aligned with a prior investigation conducted by Al-Farhan, et al in 2021, in which they made modifications to certain food options based on religious limitations and removed others that were either unavailable, not popular, or not commonly consumed among the Arab population (40).”

-Limitation of study should be in separated section.

Thank you for your comment, we have clarified it

Study strengths and limitations

“This study has two main strengths. First, it provides culturally adapted Arabic translations of the SBQ, Dietary Habits questionnaire, and the Preclinical Mobility Limitation questionnaire so that they can be widely used to assess physical behaviors in Arabic-speaking populations. Second, the translation and cultural validation process involved an adequate sample size for the cognitive interviews (50 participants), and the pre-testing sample size was adequate according to cross-cultural adaptation guidelines (32). Despite its merits, this study is subject to two limitations. First, it does not include participants <30 years old or >65; this is because these age ranges are not required by ProPASS. Second, it does not provide an examination of the psychometric properties of the current questionnaires; however, future work is expected to cover this area.”

- Implication of study should be write.

Thank you for your comments. The implication you could find it in line 412 – 41

“The SBQ, Dietary Habits questionnaire, and Preclinical Mobility Limitation questionnaire were successfully translated and culturally adapted for use with Arabic-speaking populations. These three research instruments will facilitate the collection of health data among Saudi Arabian and other Arabic-speaking populations for use with large global health initiatives such as ProPASS (10). Future studies are needed to further test the psychometric properties of these research instruments, including their test-retest reliability and validity in the Arabic context.”

---

## [Decision Letter · Decision Letter 1]

16 May 2023

Arabic Translation and Cultural Adaptation of Sedentary Behavior, Dietary Habits, and Preclinical Mobility Limitation Questionnaires: A Cognitive Interview Study

PONE-D-22-32960R1

Dear Dr. Abdulrahman,

We’re pleased to inform you that your manuscript has been judged scientifically suitable for publication and will be formally accepted for publication once it meets all outstanding technical requirements.

Kind regards,

Sally Mohammed Farghaly

Academic Editor

PLOS ONE

Additional Editor Comments (optional):

Reviewers' comments:

Reviewer's Responses to Questions

**Comments to the Author**

1. If the authors have adequately addressed your comments raised in a previous round of review and you feel that this manuscript is now acceptable for publication, you may indicate that here to bypass the “Comments to the Author” section, enter your conflict of interest statement in the “Confidential to Editor” section, and submit your "Accept" recommendation.

Reviewer #1: All comments have been addressed

Reviewer #3: (No Response)

2. Is the manuscript technically sound, and do the data support the conclusions?

Reviewer #1: Yes

Reviewer #3: Yes

3. Has the statistical analysis been performed appropriately and rigorously? 

Reviewer #1: Yes

Reviewer #3: Yes

4. Have the authors made all data underlying the findings in their manuscript fully available?

Reviewer #1: Yes

Reviewer #3: Yes

5. Is the manuscript presented in an intelligible fashion and written in standard English?

Reviewer #1: Yes

Reviewer #3: Yes

6. Review Comments to the Author

Reviewer #1: thank you so much I don't have other issues to addressed and i accept to the manuscript, and all comments was addressed

Reviewer #3: I believe this was a very well-written manuscript. Obviously it has gone through a couple of rounds of revision, and I can see that. Methods have been clearly explained, statistical analysis is demonstrated, and the discussion is meaningful. I believe this manuscript can be published as is.

7. PLOS authors have the option to publish the peer review history of their article (what does this mean?). If published, this will include your full peer review and any attached files.

Reviewer #1: **Yes: **Ayman Mohamed El-Ashry

Reviewer #3: No

---

## [Editor Report · Acceptance letter]

2 Jun 2023

PONE-D-22-32960R1 

Arabic Translation and Cultural Adaptation of Sedentary Behavior, Dietary Habits, and Preclinical Mobility Limitation Questionnaires: A Cognitive Interview Study 

Dear Dr. Alaqil:

I'm pleased to inform you that your manuscript has been deemed suitable for publication in PLOS ONE. Congratulations! Your manuscript is now with our production department. 

Kind regards, 

on behalf of

Professor Sally Mohammed Farghaly 

Academic Editor

PLOS ONE